# Under-Recognized Macrophage Activation Syndrome in Refractory Kawasaki Disease: A Wolf in Sheep’s Clothing

**DOI:** 10.3390/children9101588

**Published:** 2022-10-20

**Authors:** Sangwon Rhee, Danbi Kim, Kyoungsoon Cho, Jung Woo Rhim, Soo-Young Lee, Dae Chul Jeong

**Affiliations:** 1Department of Pediatrics, Bucheon St. Mary’s Hospital, The Catholic University of Korea, Bucheon 14647, Korea; 2Department of Pediatrics, College of Medicine, The Catholic University of Korea, Seoul 06591, Korea; 3Department of Pediatrics, Daejeon St. Mary’s Hospital, The Catholic University of Korea, Daejeon 34943, Korea; 4The Vaccine Bio Research Institute, College of Medicine, The Catholic University of Korea, Seoul 06591, Korea; 5Department of Pediatrics, Seoul St. Mary’s Hospital, The Catholic University of Korea, Seoul 06591, Korea

**Keywords:** macrophage activation syndrome, incidence, refractory Kawasaki disease

## Abstract

Recognition of macrophage activation syndrome (MAS) in patients with refractory Kawasaki disease (KD) can be challenging. This study aimed to investigate the incidence of MAS in patients with refractory KD and to compare the characteristics of refractory KD and MAS. Medical records of 468 patients diagnosed with KD from January 2010 to December 2019 were retrospectively reviewed. Of the 468 KD patients, 63 were enrolled in the study as a refractory KD group (*n* = 59) and an MAS group (*n* = 4). The incidence of MAS was 0.8% (4/468) in patients with KD and 6.3% (4/63) in patients with refractory KD. Compared to the refractory KD group, the MAS group had higher frequencies of incomplete KD, hepatosplenomegaly, third-line treatment, and MAS screening, and showed lower levels of albumin. No significant differences were found in other clinical and laboratory findings. In addition to four patients with MAS, five patients with refractory KD who received third-line treatment showed severe systemic inflammation and organ dysfunction, but only one in five patients underwent MAS screening, including ferritin levels. In conclusion, given the relatively high incidence of MAS in children with refractory KD and the similar phenotype between refractory KD and MAS, we propose that MAS screening should be included in routine laboratory tests for refractory KD.

## 1. Introduction

Kawasaki disease (KD) is an acute systemic vasculitis of unknown etiology that predominantly affects children less than five years of age [1,2]. It is characterized by the presence of prolonged fever with principal clinical features of conjunctivitis, lip redness and strawberry tongue, skin rash, changes in the extremities, and cervical lymphadenopathy [2]. Coronary artery abnormalities (CAAs) develop in about 25% of untreated children and can lead to long-term cardiovascular complications [1]. Timely initiation of treatment with intravenous immunoglobulin (IVIG) has been shown to shorten the duration of fever and reduce the prevalence of CAAs [3]. However, 10–20% of patients with KD do not respond to initial IVIG treatment and are at high risk of developing CAAs [4]. Many studies have been conducted to improve the diagnosis and treatment of refractory KD [5,6,7].

Macrophage activation syndrome (MAS), also known as secondary hemophagocytic lymphohistiocytosis (HLH), is an inflammatory phenomenon caused by excessive activation of T cells and macrophages secondary to infections, malignancies, or rheumatic diseases [8,9,10]. It is characterized by unremitting fever, hepatosplenomegaly, cytopenia, and organ dysfunction [9]. The leading causes of MAS in children are systemic juvenile idiopathic arthritis (sJIA) and systemic lupus erythematosus (SLE) [10]. With increasing reports of KD complicated by MAS, KD is considered the third most common cause of MAS in children [11]. MAS is potentially fatal, so early recognition and timely treatment are essential to achieve a good prognosis [9]. However, detection of MAS can be challenging in patients with KD, especially in patients with refractory KD [12,13,14]. Refractory KD and MAS share important clinical and laboratory characteristics, and many MAS patients exhibit IVIG resistance observed in patients with refractory KD [12]. In addition, typical MAS in patients with KD occurs as refractory KD worsens and progresses [13]. Despite the close relationship between refractory KD and MAS, only a few studies have focused on MAS developing in patients with refractory KD.

This study investigated the incidence of MAS in patients with refractory KD and compared the characteristics of refractory KD with and without MAS to find diagnostic clues for distinguishing between the two diseases.

## 2. Materials and Methods

### 2.1. Study Population and Definitions

Medical records of 468 patients who were diagnosed with KD at the Department of Pediatrics of Bucheon St. Mary’s Hospital (Bucheon, Korea), between January 2010 and December 2019, were retrospectively reviewed. All patients with KD received treatment with IVIG (2 g/kg/dose) and aspirin (30–50 mg/kg/day) and underwent echocardiography during hospitalization. Of the 468 KD patients, those who met the definition of refractory KD (i.e., the refractory KD group) and those who met the diagnostic criteria for MAS (i.e., the MAS group) were enrolled in this study. The study was approved by the Institutional Review Board of Bucheon St. Mary’s Hospital (approval number: HC22WISI0026) and informed consent was waived due to the study’s retrospective nature. 

The diagnosis of KD was based on the 2017 American Heart Association criteria [1]. Patients who presented with fever ≥5 days and ≥4 of five principal clinical features were diagnosed with complete KD and patients who present with fever ≥5 days, <4 of five principal clinical features, and compatible laboratory or echocardiographic findings were diagnosed with incomplete KD. Refractory KD was defined when KD patients had recrudescent or persistent fever ≥36 h after completion of initial IVIG treatment [5,6]. MAS was diagnosed according to the HLH-2004 criteria [8], in which KD patients met ≥5 of the following eight criteria: (1) persistent fever; (2) heptosplenomegaly; (3) cytopenia ≥ 2 cell lines (hemoglobin < 9.0 g/dL, neutrophils < 1000/μL, or platelets < 100 × 10^3^/μL); (4) hypertriglyceridemia (≥265 mg/dL) or hypofibrinogenemia (≤150 mg/dL); (5) hyperferritinemia (≥500 ng/mL); (6) hemophagocytosis in bone marrow, spleen, or lymph nodes; (7) low or absent natural killer cell activity; and (8) elevated soluble IL-2 receptor (sCD25; ≥2400 U/mL).

### 2.2. Data Collection

Clinical data were obtained for age, sex, duration of fever at diagnosis, length of hospital stay, five principal features, incomplete KD, hepatosplenomegaly, treatment, coronary artery and other complications, and outcome. Coronary arteries were categorized as abnormal if the internal lumen diameter was >3 mm in children aged <5 years and ≥4 mm in children aged ≥5 years; the internal diameter of a segment measured ≥1.5 times that of an adjacent segment; or the coronary lumen was clearly irregular [4,14]. Laboratory data on the day of admission were recorded and included hemoglobin, white blood cell (WBC) count, platelet count, erythrocyte sedimentation rate (ESR), C-reactive protein (CRP), sodium, albumin, aspartate transaminase (AST), and alanine transaminase (ALT). For patients undergoing MAS screening (i.e., ferritin, triglyceride, fibrinogen, and lactate dehydrogenase), laboratory results were separately investigated.

A literature search was performed to identify previous studies of MAS in patients with KD and refractory KD using the key terms “macrophage activation syndrome or secondary hemophagocytic lymphohistiocytosis” and “Kawasaki disease”. Studies representing the number of patients with MAS and the number of patients with KD were included and analyzed.

### 2.3. Statistical Analysis

For statistical analysis, the Fisher’s exact test was used to compare categorical data and the Mann–Whitney U test was used to compare numeric data. All tests were two-tailed, and a *p*-value < 0.05 was considered statistically significant. Statistical analyses were performed using IBM SPSS Statistics software Version 21.0 (IBM Corporation, Armonk, New York, NY, USA).

## 3. Results

### 3.1. Incidence of MAS in KD and Refractory KD

Of the 468 KD patients, 63 (13.5%) were enrolled in this study: 59 who met the definition of refractory KD and 4 who met the diagnostic criteria for MAS. All four patients diagnosed with MAS also met the definition of refractory KD. The incidence of MAS was 0.8% (4/468) in patients with KD and 6.3% (4/63) in patients with refractory KD. The clinical course of 63 patients with refractory KD after second IVIG or intravenous methylprednisolone (IVMP) treatment was as follows: 4 (6.3%) were diagnosed with MAS, 5 (7.9%) received third-line treatment, and 54 (85.7%) had no further febrile episodes (Figure 1).

### 3.2. Comparison of Characteristics between the Two Groups

The clinical and laboratory findings in the refractory KD group (*n* = 59) and the MAS group (*n* = 4) were compared (Table 1). Median age, sex ratio, duration of fever at diagnosis, length of hospital stay, and frequency of coronary artery complications were not significantly different between the two groups. The frequencies of the five principal features of conjunctivitis, lip redness and strawberry tongue, skin rash, changes in the extremities, and cervical lymphadenopathy were also not different (*p* = 0.181, *p* = 0.115, *p* = 1.000, *p* = 0.137, and *p* = 1.000, respectively). The frequencies of incomplete KD (*p* = 0.015), hepatosplenomegaly (*p* < 0.001), and third-line treatment (*p* < 0.001) were higher in the MAS group than in the refractory KD group. Comparison of laboratory findings revealed that the MAS group had lower median albumin levels (*p* < 0.004) and received MAS screening more frequently (*p* = 0.001) than the refractory KD group. There were no significant differences between the two groups in other laboratory findings.

### 3.3. Patients Who Received Third-Line Treatment

Of the enrolled patients, four (100.0%) in the MAS group and five (8.5%) in the refractory KD group received third-line treatment because of persistent fever despite second IVIG or IVMP treatment (Table 2). All four patients with MAS exhibited persistent fever, hepatosplenomegaly, elevated liver transaminases, hypertriglyceridemia or hypofibrinogenemia, and hyperferritinemia. Three patients with MAS also had cytopenia. Bone marrow biopsies were performed in three patients with MAS and of these, two had hemophagocytosis. One patient with MAS received combination chemotherapy based on the HLH-2004 therapeutic protocol and the remaining three patients with MAS received immune-modulators, such as cyclosporine, steroids, and/or additional IVIG. Three patients with MAS had organ involvement and two of them were admitted to an intensive care unit (ICU). No CAAs were observed in patients with MAS and all four patients with MAS recovered successfully.

All five patients with refractory KD who received third-line treatment showed persistent fever and elevated liver transaminases. Four patients with refractory KD had cytopenia and two of them had hepatosplenomegaly. All five patients with refractory KD received infliximab, steroids, and/or additional IVIG as third-line treatment. Four patients with refractory KD had organ involvement and of these, two were admitted to the ICU. CAAs were observed in one patient with refractory KD and all five patients with refractory KD recovered successfully. It was not possible to determine whether these five patients with refractory KD met the MAS diagnostic criteria, because MAS screening was performed in one of the five patients with refractory KD who received third-line treatment.

### 3.4. Literature Review

Six studies were analyzed that included the numbers of patients with MAS and KD (Table 3). The incidence of MAS was 0.4% to 1.9% in patients with KD and 6.7% to 7.0% in patients with refractory KD. Almost all patients with MAS (41/42) were found to be IVIG-resistant. In studies that did not represent the number of patients with refractory KD, the number of patients with refractory KD was calculated using the average proportion of refractory KD (i.e., 15% of total KD [1]). The incidence of MAS in the estimated number of patients with refractory KD was 6.5% to 12.5%.

## 4. Discussion

This study was designed to investigate the incidence of MAS in refractory KD and the characteristics of refractory KD and MAS that can be used to distinguish between the two diseases. The incidence of MAS in patients with refractory KD was 6.3% (4/63), higher than the incidence of MAS in all patients with KD (0.8%, 4/468). No specific parameters were found that could be used to distinguish between refractory KD and MAS in clinical and laboratory findings of patients with refractory KD recorded on the day of admission.

Patients with refractory KD have more severe systemic inflammation and increased risk of developing complications, including CAAs, than patients with KD [4,5,6]. Therefore, early identification of patients who do not respond to initial IVIG treatment is important in the management of KD [2,7]. Risk factors for IVIG resistance in patients with KD have been identified as prolonged fever, young age, leukocytosis, thrombocytopenia, high CRP levels, and/or elevated liver transaminases [5,6,7]. These risk factors are useful for predicting IVIG resistance, but they are not useful for distinguishing refractory KD and MAS because they are found in both refractory KD and MAS patients [12,18]. The mortality rate of MAS in patients with KD was reported to be 13–25% [13,20]. To not overlook MAS, it is necessary to consider the possibility of developing MAS in patients with KD, especially those with risk factors for IVIG resistance.

Since Ohga et al. [22] first reported the co-occurrence of KD and MAS in 1995, more than 50 articles have been published on MAS in patients with KD [23]. The well-known incidence of MAS in patients with KD is 1.1–1.9% [16,17]. The incidence of MAS in severe forms of KD is expected to be higher than in typical KD. In fact, the incidence of MAS was reported to be 4.8% (3/63) in KD shock syndrome [24] and 3.2% (8/247) in incomplete KD [14], both of which have more severe clinical manifestations than typical KD. In this study, the incidence of MAS in refractory KD was 6.3%, which is similar to that found in previous studies by Roh et al. [20] and Qiu et al. [21] (Table 3). The reported incidence of MAS is 7.0–13.0% in children with sJIA and 0.9–4.6% in children with SLE [9,25]. Combining the data with findings from the present study, the incidence of MAS in children should be rearranged in the following order: sJIA (~13%) > refractory KD (~7%) > SLE (~5%) > typical KD (~2%).

In children with sJIA, occult MAS is reported to be at least three times more common than overt MAS [9,10]. sJIA patients with occult MAS show clinical manifestations that do not currently meet the diagnostic criteria for MAS [13,26]. However, their clinical manifestations can worsen and eventually progress to overt MAS [27]. Therefore, when a patient with sJIA shows clinical exacerbation despite proper treatment, the patient is defined as having occult MAS that can progress to overt MAS [26,27]. Similarly, when a patient with KD shows persistent fever despite IVIG and/or IVMP treatment (i.e., refractory KD), the patient should be considered occult MAS and MAS screening should be performed [28].

MAS can be under-recognized in patients with KD and refractory KD for several reasons. First, KD and MAS share many clinical and laboratory findings [12,18]. In actual practice, the shared phenotypes between KD and MAS, such as fever, rash, and elevated liver transaminases, can lead to diagnostic confusion [16,17]. Distinguishing MAS from refractory KD is more difficult than distinguishing MAS from KD because refractory KD has more MAS-like phenotypes than KD [13]. For example, hepatosplenomegaly is very rare in patients with KD but can be seen in patients with refractory KD [29]. Dyslipidemia, including hypertriglyceridemia, also occurs more frequently in refractory KD than in KD [30]. Second, the lack of awareness of the possible co-occurrence of KD and MAS is a more fundamental reason than others [12,21]. In this study, only 15% (9/59) of the refractory KD group underwent MAS screening (i.e., ferritin, triglyceride, fibrinogen, and lactate dehydrogenase). Pilania et al. [19] have highlighted that the application of MAS screening to patients with KD was determined by physician experience rather than by clinical severity. Given the low incidence of MAS in KD (0.4–1.9%), it may be not necessary to include MAS screening in routine laboratory tests for KD [20]. However, given the relatively high incidence of MAS in refractory KD (6.3–7.0%), it is worthwhile to include MAS screening in routine laboratory tests for refractory KD. Thirdly, the absence of specific diagnostic criteria is one possible explanation for under-recognized MAS in KD and refractory KD [12,13]. Most studies, including the present one, have adopted the HLH-2004 criteria to diagnose MAS in patients with KD [23]. While the HLH-2004 criteria are useful, they are not sensitive for diagnosing MAS in sJIA or KD because they were originally developed for primary HLH [9,11]. To address these limitations, the 2016 consensus criteria for MAS in patients with sJIA are proposed [31], which also suggested an application for MAS in patients with KD [32]. García-Pavón et al. [23] found that the 2016 criteria for MAS in patients with sJIA are highly sensitive and specific in patients with KD compared to the HLH-2004 criteria. Well-designed clinical studies are needed to determine the diagnostic criteria with appropriate sensitivity and specificity for MAS in patients with KD.

Considering the reasons for under-recognized MAS in KD described above, this study identified five patients with refractory KD who received third-line treatment because of persistent fever despite second IVIG or IVMP treatment (Table 2). They showed severe clinical manifestations similar to MAS, such as cytopenia, organ dysfunction, and/or ICU admission. However, only one of these five patients underwent MAS screening due to the lack of awareness of the co-occurrence of KD and MAS. In other words, MAS may not have been recognized in these five patients with refractory KD. 

This study had limitations: it was small-sized, retrospective, and conducted in a single center. Large-scale clinical studies are needed to support study findings. All enrolled patients had routine laboratory tests for KD, but only one-fifth of enrolled patients had additional laboratory tests for MAS. This is an important finding as well as a major limitation of this study.

## 5. Conclusions

When refractory KD is considered a disease entity, it is the second most common cause (~7%) of MAS in children after sJIA (~13%). However, the recognition of MAS may be more difficult in patients with refractory KD than in patients with KD because refractory KD has more MAS-like features. Given the relatively high incidence of MAS in children with refractory KD and the possibility of under-recognized MAS in refractory KD, we propose that MAS screening, including ferritin levels, should be included in the routine laboratory tests performed for refractory KD.

## Figures and Tables

**Figure 1 children-09-01588-f001:**
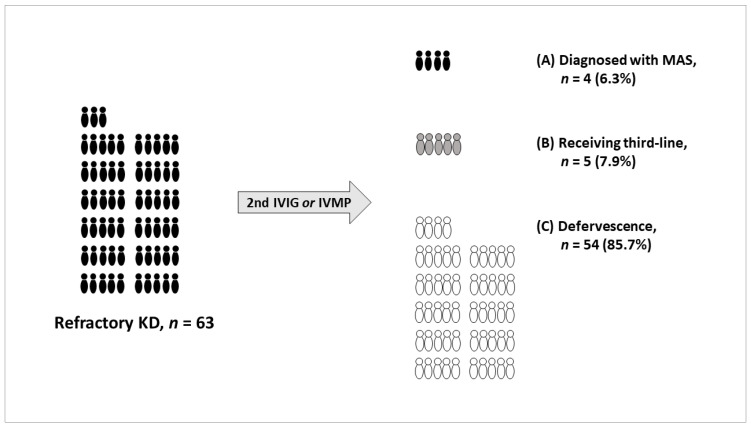
Clinical course of 63 patients with refractory KD after second IVIG or IVMP treatment. (A) Four patients (6.3%) diagnosed with MAS, (B) five patients (7.9%) receiving third-line treatment, and (C) fifty-four patients (85.7%) with no further febrile episodes. KD, Kawasaki disease; MAS, macrophage activation syndrome; IVIG, intravenous immunoglobulin; IVMP, intravenous methylprednisolone. Third-line treatment refers to cyclosporine, steroids, HLH-2004 therapeutic regimen [8], intravenous methylprednisolone, intravenous immunoglobulin, or infliximab.

**Table 1 children-09-01588-t001:** Comparison of clinical and laboratory findings between groups.

	Refractory KD (*n* = 59)	MAS (*n* = 4)	*p*-Value
Clinical			
Age (years)	2.6 (0.3–11.1)	4.4 (1.6–13.1)	0.481
Sex (male)	38 (64.4)	3 (75.0)	1.000
Duration of fever (days)	4.0 (2.0–10.0)	5.0 (4.0–7.0)	0.087
Length of hospital stay (days)	9.0 (3.0–24.0)	12.0 (10.0–41.0)	0.064
Principal features (e.g., conjunctivitis) ^a^	57 (96.6)	3 (75.0)	0.181
Incomplete KD	8 (13.6)	3 (75.0)	0.015
Hepatosplenomegaly	2 (3.4)	4 (100.0)	<0.001
Third-line treatment ^b^	5 (8.5)	4 (100.0)	<0.001
Coronary artery complications	10 (16.9)	0 (0.0)	1.000
Laboratory			
Hemoglobin (g/dL)	11.4 (8.8–13.7)	10.0 (5.7–11.6)	0.108
WBC count (×10^9^/L)	16.3 (2.7–49.3)	8.0 (2.1–32.8)	0.210
Platelet count (×10^9^/L)	310.0 (26.0–545.0)	68.5 (59.0–414.0)	0.071
ESR (mm/h)	58.0 (3.0–120.0)	39.0 (12.0–62.0)	0.143
CRP (mg/dL)	9.9 (0.8–25.9)	18.5 (7.8–21.5)	0.055
Sodium (mmol/L)	136.0 (127.0–141.0)	135.0 (134.0–135.0)	0.243
Albumin (g/dL)	3.6 (2.1–4.8)	2.7 (2.5–2.9)	0.004
AST (U/L)	76.0 (18.0–754.0)	234.0 (75.0–964.0)	0.065
ALT (U/L)	85.0 (9.0–498.0)	233.0 (26.0–1099.0)	0.190
MAS screening ^c^	9 (15.3)	4 (100.0)	0.001

Values are presented as numbers (%) or median and range. KD, Kawasaki disease; MAS-KD, macrophage activation syndrome complicating KD; WBC, white blood cell; ESR, erythrocyte sedimentation rate; CRP, C-reactive protein; AST, aspartate transaminase; ALT, alanine transaminase. ^a^ The frequencies of the five principal features were not statistically different between the two groups (see text). ^b^ Third-line treatment refers to cyclosporine, steroids, HLH-2004 therapeutic regimen [8], intravenous methylprednisolone, intravenous immunoglobulin, or infliximab. ^c^ MAS screening includes ferritin, triglyceride, fibrinogen, and lactate dehydrogenase.

**Table 2 children-09-01588-t002:** Clinical and laboratory characteristics of patients who received third-line treatment.

	MAS (*n* = 4)				Refractory KD (*n* = 5)				
	1	2 ^a^	3	4	1	2	3	4	5
Sex/age (years)	M/1.8	F/1.6	M/7.0	M/13.1	F/2.3	F/0.5	M/6.1	M/2.5	M/3.0
Fever despite 2nd IVIG or IVMP	Yes	Yes	Yes	Yes	Yes	Yes	Yes	Yes	Yes
Hepatosplenomegaly	Yes	Yes	Yes	Yes	No	No	Yes	Yes	No
Cytopenia (≥2 cell lines)	No	Bicyto-	Pancyto-	Bicyto-	No	Anemia	Thrombo-	Anemia	Thrombo-
AST/ALT (U/L) ^b^	190/236	964/1099	278/230	75/26	189/374	146/86	299/181	137/338	48/19
Abnormal TG or fibrinogen	Yes	Yes	Yes	Yes	–	–	–	–	No
Ferritin (>500 ng/mL)	5130	57100	1420	790	–	–	–	–	570
Hemophagocytosis	Yes	Yes	No	–	–	–	–	–	–
Management and outcomes									
Admission to ICU	Yes	No	No	Yes	No	No	Yes	Yes	No
Third-line treatment	IVMP	CS + DX	HLH-2004	IVIG + DX	IVMP	IVMP	IVIG + IVMP	IVMP	Infliximab
Coronary artery complications	No	No	No	No	No	Yes	No	No	No
Other complications	Shock	Arthritis	No	DIC, seizure	GB hydrops	Meningitis	ARDS, shock	AKI, seizure	No
Successful recovery	Yes	Yes	Yes	Yes	Yes	Yes	Yes	Yes	Yes

KD, Kawasaki disease; MAS, macrophage activation syndrome; –, not available; IVIG, intravenous immunoglobulin; IVMP, intravenous methylprednisolone; bicyto-, bicytopenia; pancyto-, pancytopenia; thrombo-, thrombocytopenia; AST, aspartate transaminase; ALT, alanine transaminase; TG, triglyceride; ICU, intensive care unit; CS, cyclosporine; DX, dexamethasone; HLH, hemophagocytic lymphohistiocytosis, DIC, disseminated intravascular coagulation; GB, gallbladder; ARDS, acute respiratory distress syndrome; AKI, acute kidney injury. ^a^ Patient 2 was reported in our previous study [15]. ^b^ Elevated liver transaminases are used as supportive evidence of MAS but are not included in the HLH-2004 diagnostic criteria.

**Table 3 children-09-01588-t003:** Literature review of the incidence of MAS in KD and refractory KD [16,17,18,19,20,21].

	Subjects	MAS in KD	IVIG Resistance in MAS	MAS in Refractory KD
Latino et al. [16], Canada	*n* = 638	12/638 (1.9%)	12/12 (100.0%)	12/96 (12.5%) ^b^
Wang et al. [17], China	*n* = 719	8/719 (1.1%)	7/8 (87.5%)	7/108 (6.5%) ^b^
Mousavi et al. [18], Iran	*n* = 218	4/218 (1.8%)	4/4 (100.0%)	4/33 (12.1%) ^b^
Pilania et al. [19], India	*n* = 950	12/950 (1.3%)	12/12 (100.0%)	12/143 (8.4%) ^b^
Roh et al. [20], Korea	*n* = 158	5/158 (3.2%) ^a^	5/5 (100.0%)	5/71 (7.0%)
Qiu et al. [21], China	*n* = 244	1/244 (0.4%)	1/1 (100.0%)	1/15 (6.7%)
This study, Korea	*n* = 468	4/468 (0.8%)	4/4 (100.0%)	4/63 (6.3%)

MAS, macrophage activation syndrome; KD, Kawasaki disease; IVIG, intravenous immunoglobulin. ^a^ The incidence of MAS in KD is exceptionally high, probably because subjects with severe clinical manifestations and who were tested for serum ferritin levels were included in their prospective study [20]. ^b^ The number of patients with refractory KD was calculated using the average proportion of refractory KD (i.e., 15% of total KD [1]) in studies that did not represent the number of patients with refractory KD.

## Data Availability

Data from this study can be obtained by request to the authors.

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
