# Peer review of "Under-Recognized Macrophage Activation Syndrome in Refractory Kawasaki Disease: A Wolf in Sheep’s Clothing"

_children, 2022, doi:10.3390/children9101588_

Round 1

Reviewer 1 Report

Rhee and colleagues report on a cohort of children with KD who were refractory to IVIg and a subset with MAS. There is not much new here. They likely underdiagnosed MAS as HLH-04 criteria are too strict (designed for familial HLH) and inappropriate for diagnosing secondary (including rheumatic) HLH/MAS. The manuscript will benefit by addressing the following questions/comments/concerns:

1. The 2016 Ravelli sJIA/MAS criteria should be explored for diagnosing MAS among this cohort.

2. Why was anakinra not used for refractory KD?

3. How was completeness of diagnosis observed? Were the patients sequentially identified in the time-frame by using ICD10 codes in the electronic medical record?

Line 57, would be clearer if written as, "... compared the characteristics of refractory KD with and without MAS to find ..."

Line 54, "developed" should be "developing"

Author Response

Point 1: The 2016 Ravelli sJIA/MAS criteria should be explored for diagnosing MAS among this cohort.

Response 1: I agree with you. I commented in a previous study that the 2016 Ravelli sJIA/MAS criteria should be applied in children with KD as well as with sJIA (Ann. Rheum. Dis. 2016, 75, e44, Reference 32).

All four KD patients with MAS in our study met both the 2016 Ravelli sJIA/MAS criteria and the HLH-2004 criteria. The 2016 Ravelli sJIA/MAS criteria are highly sensitive and specific in patients with KD compared to the HLH-2004 criteria. I emphasized the need for the application of the 2016 criteria in patients with KD in the ‘Discussion’ session with new reference.

Point 2: Why was anakinra not used for refractory KD?

Response 2: The attached figure shows the practice recommendation for sHLH/MAS, as applied in my hospital.

Anakinra is the first choice for sHLH/MAS refractory to IVMP and cyclosporine. Tocilizumab may be used if ankinra is not available.

The problem is that ankinra is not commercially available in Korea. For medical condition that treatment of ankinra is essential, the patient’s parents have to visit National Orphan Drug Center.    

Last year, anakinra was administered to a patient with multisystem inflammatory system in children (MIS-C) at our hospital. At that time, the parents visited National Orphan Drug Center. Recently, we had a patient with sJIA and MAS. The patient received tocilizumab for refractory MAS.

Point 3: How was completeness of diagnosis observed? Were the patients sequentially identified in the time-frame by using ICD10 codes in the electronic medical record?

Response 3: My university has a special program for the clinical research called the Clinical Data Warehouse (CDW), a warehouse that integrates clinical data such as de-identified patient diagnoses, prescriptions, test results, and health information). I can use this program with the approval of the Institutional Review Board (IRB).

Briefly, the parameter “M303 (ICD10) + IVIG/aspirin treatment, over the past 10 years” was used.

Point 4: Line 57, would be clearer if written as, "... compared the characteristics of refractory KD with and without MAS to find ..."

Response 4: The sentence has been modified as you pointed out.

Point 5: Line 54, "developed" should be "developing"

Response 5: The sentence has been modified as you pointed out.

Reviewer 2 Report

The authors sought to investigate the incidence of MAS among a retrospective cohort of patients diagnosed with KD. MAS was documented in 6.3% of refractory KD patients and 0.8% of the whole cohort. The study is well-written and informative. 

I suggest specifying what “MAS screening” stands for and highlighting the importance of looking for MAS in KD patients unresponsive to the first cycle of IVIG. 

Please explain the sentence “In studies that did not represent the number of patients with refractory KD, the estimated incidence of MAS, calculated using the average proportion of refractory KD (i.e., 15% of total KD [1]), was 6.5% to 12.5% in the patients with refractory KD”.

Please add references line 214

Author Response

Point 1: I suggest specifying what “MAS screening” stands for and highlighting the importance of looking for MAS in KD patients unresponsive to the first cycle of IVIG.

Response 1:

1-1. What “MAS screening” stands for

MAS screening includes ferritin, triglyceride, fibrinogen, and lactate dehydrogenase.

The detailed components of MAS screening were described in the ‘Methods’ and ‘Discussion’ sections. In addition, “MAS screening, including ferritin levels” has been added to the ‘Abstract’ and ‘Conclusions’ sections.

1-2. The importance of looking for MAS in KD patients unresponsive to the first cycle of IVIG.

To avoid overlooking MAS in patients with KD, it is helpful to consider refractory KD (i.e., unresponsive to the first cycle of IVIG) as occult MAS that can progress to overt MAS.

I’ve added this point to the ‘Discussion’ section.

“In children with sJIA, occult MAS is reported to be at least three times more common than overt MAS [9,10]. sJIA patients with occult MAS show clinical manifestations that do not currently meet the diagnostic criteria for MAS [13,26]. However, their clinical manifestations can worsen and eventually progress to overt MAS [27]. Therefore, when a patient with sJIA shows clinical exacerbation despite proper treatment, the patient is defined as having occult MAS that can progress to overt MAS [26,27]. Similarly, when a patient with KD shows persistent fever despite IVIG and/or IVMP treatment (i.e., refractory KD), the patient should be considered occult MAS and MAS screening should be performed [28].”

Point 2: Please explain the sentence “In studies that did not represent the number of patients with refractory KD, the estimated incidence of MAS, calculated using the average proportion of refractory KD (i.e., 15% of total KD [1]), was 6.5% to 12.5% in the patients with refractory KD”.

Response 2: The sentence has been modified as follows.

“In studies that did not represent the number of patients with refractory KD, the number of patients with refractory KD was calculated using the average proportion of refractory KD (i.e., 15% of total KD [1]). The incidence of MAS in the estimated number of patients with refractory KD was 6.5% to 12.5%.”

Point 3: Please add references line 214

Response 3: References 20 and 21 were added

Round 2

Reviewer 1 Report

The authors have adequately addressed the reviewers' concerns.